# Video Affective Impact Prediction with Multimodal Fusion and Long-short Temporal Context

## Abstract

Predicting the emotional impact of videos using machine learning is a challenging task. Feature extraction, multi-modal fusion and temporal context fusion are crucial stages for predicting valence and arousal values in the emotional impact, but have not been successfully exploited. In this paper, we proposed a comprehensive framework with innovative designs of model structure and multi-modal fusion strategy. We select the most suitable modalities for valence and arousal tasks respectively and each modal feature is extracted using the modality-specific pre-trained deep model on large generic dataset. Two-time-scale structures, one for the intra-clip and the other for the inter-clip, are proposed to capture the temporal dependency of video content and emotional states. To combine the complementary information from multiple modalities, an effective and efficient residual-based progressive training strategy is proposed. Each modality is step-wisely combined into the multi-modal model, responsible for completing the missing parts of features. With all those above, our proposed prediction framework achieves better performance with a large margin compared to the state-of-the-art.

## 1 Introduction

Affective video content analysis aims at predicting the videos' emotional impact on audiences. It plays important roles in understanding the videos' content, highlight detection, and of a fundamental support for several advanced applications such as multi-modal search with sentimental queries. Predicting the audiences' emotional evolvement when watching movies is also an important way to help both online media-server providers or filmmakers to invest movies, evaluate on-line effect as well as distribute them more efficiently.

In the affective computing community, human emotions can be categorically or continuously defined (Izard, 2007; Barrett et al., 2007). Emotion categories are happy, sad, angry, surprise, disgust, neutral, which are commonly used in emotion recognition or classification (Cowie et al., 2001). Compared to the categorical definition, continuous definition describes the emotions continuously in two dimensions: Valence (positive vs. negative) and Arousal (active vs. calm). Any human emotion can be located in the space spanned by the two dimensions, which is more fine-grained than the categorical definition. The goal of our task is to predict the audiences' emotional states based on the movie content, i.e. the valence and arousal values with the movie going on.

Finding discriminative features from raw videos for predicting valence and arousal values is far away from an easy task. Video is the typical multi-modal media involving both audio and visual modalities. Even if in visual content, human facial expressions, pose behaviors, scenes, etc. can also be regarded as modalities. Audiences' emotions can be triggered by any modality such as the actors' expressions or actions, the movies' scenes (environment, atmosphere) as well as background music. Therefore, the mainstream of affective video content analysis is to extract multi-modal features and combine all those features.

Feature fusion is another challenging step. Multi-modal features are always complementary and the importance of each modality dynamically changes over time. For example, some movie clips' emotional impact can be captured by audio content while others may rely on visual features. Current studies of affective video content analysis mainly adopt either decision-level fusion (Dobrišek

et al., 2013) or feature-level fusion (Wimmer et al., 2008). The former combines results from each modality through voting or weighted average methods. Each modality-specific model is trained independently which can't exploit the complementary information between modalities. The latter concatenates multi-modal features and learn parameters for all modalities at the same time which can easily lead to overfitting. We design a progressive training algorithm where each modality is trained and fine-tuned stepwisely. Each modality is only responsible for completing the missing parts of features extracted from previous modalities, thus the most discriminative modalities can be dynamically selected for each movie clip and the complementarity of multi-modal features can be fully utilized. The overfitting risk can also be suppressed since fewer parameters are learned at each step.

Besides all the above, we also investigate how to utilize the temporal context of videos for sentiment prediction, which is lacking in most of the related works, where they simply apply LSTMs (Hochreiter & Schmidhuber, 1997) or GRUs (Cho et al., 2014) for temporal dependency. We propose two-time-scale model structures considering the video's long-short temporal context. For the short-time context, LSTMS are used for each modality. For the long-time dependency of the valence task, a structure that is similar to the temporal segment network (TSN) (Wang et al., 2016) is used to capture the long temporal context. For the arousal task, a moving mean post-processing method is adopted to utilize the trend of previous emotions.

The contributions of this paper are as follows: We propose an effective multi-modal fusion network and design a two-time-scale model structure considering the video's long-short temporal context for affective impact prediction. Also, a residual-based progressive training strategy is used to train the fusion network to fully utilize the complementary and representative capability of each modality. Our model and training strategy achieve a new state-of-art on several related tasks.

## 2 RELATED WORKS

Video content involves both audio and visual elements. The basic part of affective video content analysis involves extracting audio and visual features to characterize the video content. In early stages, most works (Xu et al., 2013; Moreira et al., 2015) extract handcrafted features such as Local Binary Patterns (LBP), Histogram of oriented gradient (HOG), etc, to represent visual features, and Linear Predictive Coding coefficients (LPC), Mel Frequency Cepstral Coefficient (MFCC) to represent audio features. In recent years, with the development of deep learning, semantics-meaningful features extracted by deep models are becoming more and more popular in affective video content analysis and many works extract features with pre-trained CNN models. For example, A VGG-like model named VGGish (Hershey et al., 2017) is adopted for extracting audio features and CNN models trained on generic task datasets, such as ImageNet (Deng et al., 2009), RAF (Li et al., 2017) are used for extracting visual features (Liu et al., 2018).

Another focus of the emotion prediction is multi-modal fusion (Atrey et al., 2010). Multi-modal fusion methods can be divided into two categories: feature-level fusion, and decision-level fusion. The key difference between the above two categories is the stage when the fusion happens. Rosas et al. (2013) concatenate linguistic, audio and visual features into a common feature vector and seek to find the hyperplane that best separates positive examples from negative examples using SVMs (Cortes & Vapnik, 1995) with linear kernels. Wang & Cheong (2006) characterized every scene through concatenating audio and visual features to a vector and adopt a specially adapted variant of SVM to recognize anger, sadness, surprise, happiness, disgust and neutral. Metallinou et al. (2010) model face, voice and head movement cues for emotion recognition and fuse the results of all classifiers using a Bayesian framework. While those models can outperform unimodal models, they fail to utilize the dependencies among different modalities. Pang et al. (2015) use Deep Boltzmann Machine (DBM) to learn the highly non-linear relationships that exist among low-level features across different modalities for emotion prediction. Gan et al. (2017) propose a multi-modal deep regression Bayesian network (MMDRBN) to capture the dependencies between visual elements and audio elements and a fast learning algorithm is designed to learn the regression Bayesian network (RBN), Then the MMDRBN is transformed into an inference network by minimizing the KL-divergence.

To utilize the temporal context, Kurpukdee et al. (2017) extract phoneme-based features from raw input speech signals using convolutional long short-term memory (LSTM), recurrent neural network (ConvLSTM-RNN) and adopt support vector machines (SVM) or linear discriminant analysis

(LDA) to classify four emotions (anger, happiness, sadness and neutral). Fan et al. (2016) use RNNs to fuse features extracted by the convolutional neutral network (CNN) over individual video frames and use C3D (Tran et al., 2015) to encode appearance and motion information at the same time, then the predicted scores from different models are combined in a weighted-sum rule.

# 3 PROPOSED METHODS

The overall framework consists of three major steps as shown in Fig 1. We divide the untrimmed movies to non-overlap short clips with same length and predict valence and arousal for each clip. First, we extract frame-level modality-specific features using commonly-used pre-trained audio and image feature extractors. Then, the intra-clip feature fusion and inter-clip context fusion are used to consider the short and long temporal context. For intra-clip feature fusion, we append LSTMs after the frame-level feature extractor to consider the short-time temporal relation within the clip for each modality. Then, all modality-specific features are summed into a vector, representing the clip-level multi-modal features. For the inter-clip context, we adopt LSTMs based on the clip-level features to capture the long-time dependency between clips for the valence task. For the arousal task, a more simple but effective exponential moving average with decay weights is utilized. A residual-based progressive training algorithm is designed to train the network.

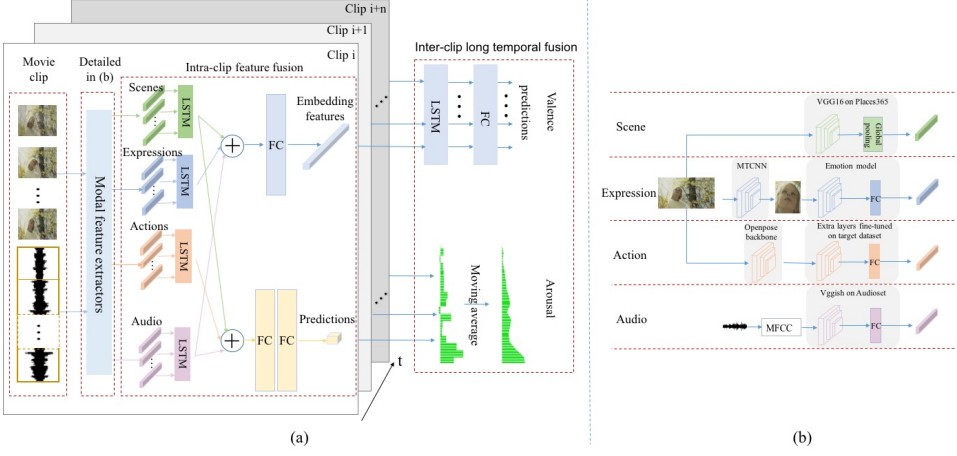

Figure 1: Part (a) describes our overall framework for valence and arousal tasks separately. Features for multi-modal are extracted as detailed in part (b)

## 3.1 MULTI-MODAL FEATURES

The modal scope we could use for valence and arousal tasks in movie affective analysis is based on the following observations. 1) The actors/Actress' actions, dialogues, and facial expressions are the key factors that affect the audience's emotions. Scenes (such as environment, atmosphere) and background music also implicitly deliver emotions to audiences. 2) Valence and Arousal emotional states depend on different modalities. For instance, Facial expressions are more related to the valence task while actions are more related to the arousal task. The actors' actions can affect the intensity of emotions while they are less related to the direction of emotions (Detenber & Reeves, 1996). Therefore, for the valence task, audio, scene and facial expression modalities are finally used. For the arousal task, audio, scene, and action modalities are finally used. Related experiments in section 4.2.1 and 4.2.2 also support our observations.

### 3.1.1 AUDIO FEATURES

Audio information in a movie can be divided into two main parts: the physical characteristics of sound and the content of language. We only focus on language-independent audio features. Here we adopt VGGish to extract semantically meaningful features with all audio characteristics taken into

consideration. The inputs are log Mel-spectrograms with shape 96x64 computed with 0.96s long audio clips. We adopt the first 960 ms from every second of the audio. We adopt the output of the 128-wide fully connected layer followed by a PCA transformation and quantization as the compact embedding features for audio. Thus, every second of audio is converted into a 128-dimension vector.

### 3.1.2 VISUAL FEATURES

Visual features consist of the human' actions, expressions as well as scenes in videos. A sparse sampling strategy, one frame per second, is adopted to sample frames from clips. Each feature will be extracted in frame-level.

We adopt pose-related features to represent the human' actions. Specifically, we append two groups of convolution/maxpool layers and a 128-wide fully connected layer after the last convolution layer of the pre-trained OpenPose's backbone (Cao et al., 2018). The weights of the OpenPose's backbone are fixed and the newly-added layers will be fine-tuned on the target dataset. To represent the actors' expressions, we pre-train an Xception network (Chollet, 2017) with fully connected layers on RAF to extract emotional facial features. To extract facial expression feature, we first detect faces from frames by MTCNN (Zhang et al., 2016). Then we crop the largest face detected and resize it to 160x160. The face image is fed into the pre-trained model and is represented by a 3072-dimension vector extracted from the last fully connected layer. If no faces are detected, the average face across the whole training dataset is used. A pre-trained VGG16 network on Places365 is adopted to extract features of the movies' scenes. We resize the frames to 224x224 and extract 512-dimension features at the last pooling layer.

## 3.2 FEATURE FUSION WITH LONG-SHORT TEMPORAL CONTEXT

### 3.2.1 INTRA-CLIP SHORT TEMPORAL FUSION

Since each modality has its own time-dependency, we first carry out short-temporal fusion for each modality. After the frame-level features for each modality are obtained, we adopt two layers of bidirectional LSTMs for each modality inside the clip to fuse the temporal information. The hidden state of the final step of the LSTMs is adopted as the clip-level modality-specific features. Having obtained the modality-specific features, we sum all the features together for each clip. Then a fully connected layer is followed, forming the final clip-level multi-modal features.

### 3.2.2 INTER-CLIP LONG TEMPORAL FUSION

We adopt two different methods to utilize the temporal context among clips for valence and arousal tasks. For the valence task, we adopt a TSN-like structure to utilize the temporal context among continuous clips. Specifically, we use two-layers bidirectional LSTMs to combine the temporal context among continuous clips. The input for each step is the clip-level multi-modal features obtained in the intra-clip short temporal fusion. The hidden state for each step is the final embedding features combining long temporal context for the corresponding clip. A fully connected layer with Tanh activation is adopted to make a final valence value prediction for each clip.

For arousal value prediction, we directly append one fully connected layer with Tanh activation as the raw arousal prediction after the intra-clip features. Then a simple but effective exponential moving average with decay weights is utilized to consider the temporal context. The reason we use moving average instead of LSTMs like the valence task is that the arousal value represents the intensity of emotions, which cannot dramatically change in relative short time, while for valence, the emotion direction might be more context-related.

## 3.3 TRAINING STRATEGY

Training the whole network, which has multiple stages with several sub-structures, in an end-to-end way is not an easy task, which can easily lead to be over-fitting because of the large number parameters. Thus, we train the whole network part by part. For VGGish, facial expression model, VGG16 pre-trained on Places365, and the modified OpenPose model, the weights are always fixed in our tasks using the pre-trained weights. For the LSTMs that and feature fusion network, we use a residual-based progressive training strategy.

### 3.3.1 RESIDUAL-BASED PROGRESSIVE TRAINING STRATEGY

To learn the LSTMs for every modality in short temporal fusion and effectively fuse multi-modal features, we design a two-stage residual based progressive training strategy as shown in Fig 2.

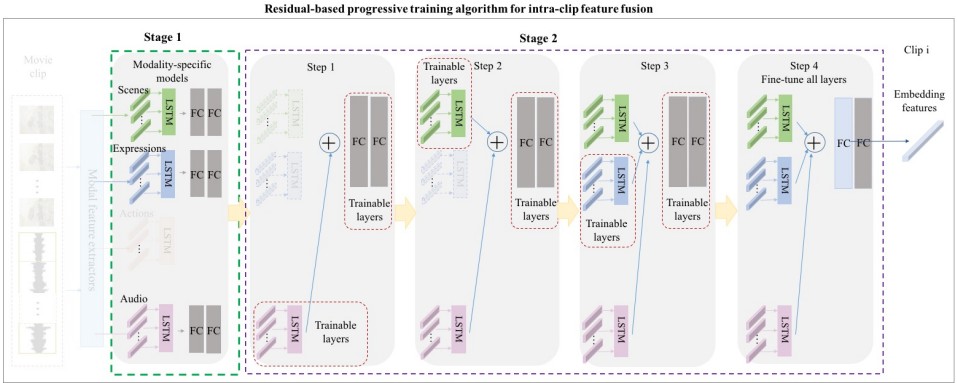

Figure 2: Residual-based progressive training strategy for intra-clip feature fusion of valence

*Stage 1: Modality-specific pre-training*

The main role of this step is to obtain the importance of each modality and determine the training order for each modality in the next stage. We append two fully connected layers for auxiliary training after the LSTMs of each modality to form the modality-specific model. Then we train each modality-specific model by minimizing the Mean Square Error (MSE) loss. The loss function can be computed as follows:

$$L(y, G) = \sum_{j=1}^{m} \frac{1}{m} ||y_j - G_j||_2^2 \tag{1}$$

Where $y_j$ is the prediction for clip $j$. $G_j$ represents the ground truth for clip $j$. It is the mean of labels for every second within the clip. $m$ denotes the batch size.

After the models are trained, we sort modalities in descending order according to their performance. Thus, modality $i$ means the modality with the $ith$ high performance and the LSTMs for it will be trained in sequence of $i$.

*Stage 2: Training intra-clip model*

At the first step, we train the LSTMs for modality 1. At the second step, we sum the features extracted by the LSTMs for modality 1 and modality 2 and only train the LSTMs for modality 2. The remaining modalities' LSTMs are learned similarly. At each step, two new fully connected layers are appended after the summed features to make predictions. Note that all the fully connected layers are only used for auxiliary LSTMs pre-training. After we get the weights for all modalities' LSTMs, we add two fully connected layers after the LSTMs and fine-tune the entire model. For the valence task, the output of the first fully connected layer is used as the clip-level embedding features. For the arousal task, the fully connected layers are used to make the prediction for each clip. The training process for the valence task is shown in Fig 2 as an example.

With the residual-based progressive training strategy, the model can dynamically select important modalities. Formally, assuming that we are training the $ith$ modality, the features combining the first $i-1$ modalities are denoted as $f_{i-1}$. Thus the LSTMs for modality $i$ fit the mapping $H_i(x) := f_i - f_{i-1}$. If $f_{i-1}$ is discriminative enough to make correct predictions and modality $i$ is of low importance, it would push the mapping to zero. If modality $i$ plays an important role, it would push the mapping to complete the $f_{i-1}$ towards $f_i$ to get better performance.

### 3.3.2 Long Temporal Fusion

For arousal, there is no parameters to train for long temporal fusion since we adopt an exponential moving average post-processing method over the predictions made for intra-clip part. Formally, the final prediction can be computed as follows:

$$ema_i = \beta * ema_{i-1} + (1 - \beta) * y_i \tag{2}$$

where $y_i$ is the prediction for clip $i$, $ema_i$ denotes the exponential moving average value for clip $i$ which is used as the final prediction, $\beta$ is the weight decay.

For valence, we keep the weights for the intra-clip part fixed and only train the LSTMs and fully connected layer for the inter-clip part. The loss function can be computed as follows:

$$L(Y, G) = \sum_{j=1}^{m} \frac{1}{m * L} \sum_{i=1}^{L} ||y_j^i - G_j^i||_2^2 \tag{3}$$

Where $y_j^i$ and $G_j$ denote the prediction and ground truth for the $ith$ clip in the $jth$ example respectively. $m$ represents the batch size. $L$ is the number of clips in an example.

## 4 Experiments and Results

### 4.1 Dataset and Metrics

The LIRIS-ACCEDE dataset is the largest dataset for affective video content analysis, which is used in the MediaEval 2018 emotional impact of movies task. The LIRIS-ACCEDE dataset contains videos from a set of 160 professionally made and amateur movies. Several movie genres are represented in this collection of movies such as horror, comedy, drama, action and so on. Languages are mainly English with a small set of Italian, Spanish, French, and others. A total of 54 movies (total duration of 26 hours and 49 minutes) from the set of 160 movies are provided as the development set. 12 other movies (total duration of 8 hours and 56 minutes) consist the test set. The scores of valence and arousal which range from -1 to 1 are provided continuously (every second) along movies. Valence is defined on a continuous scale from most negative to most positive emotions, while arousal is defined continuously from calmest to most active emotions. The official metric is the Mean Square Error (MSE), which is the common measure generally used to evaluate regression models. However, we also consider Pearson's Correlation Coefficient (PCC) for the emotional trend analysis of movies.

### 4.2 Results

#### 4.2.1 Modality-specific Performance

To evaluate the importance of every modality for each task, we train a set of modality-specific models, i.e. only using one modality in the overall prediction framework. The performance for modality-specific model is shown in Table 1. To better demonstrate each modality's effectiveness, we visualize the features representations of each modality in the test set. T-SNE is adopted for dimensionality reduction.

As shown in Table 1, the modality-specific model trained on audio features has the best performance for both valence and arousal tasks. This demonstrates that audio signals contain more emotional information. For the valence task, expressions and scenes play similar importance while actions have the worst performance. For the arousal task, actions and scenes have equal status while expressions are useless. The importance of each modality can also be revealed by the discriminative of features extracted through the modality-specific models as shown in Fig 3. This demonstrates our analysis in section 3.1.

#### 4.2.2 Multi-Modal Performance for Intra-clip Part

Here, we compare the performance of different multi-modal combinations and fusion strategies in intra-clip prediction. According to the analysis and experiments, audio and scenes are used for both

Table 1: Performance for each modality-specific models.

| Modality used | Valence | | Arousal | |
|---|---|---|---|---|
| | MSE | PCC | MSE | PCC |
| Audio | 0.098 | 0.264 | 0.140 | 0.172 |
| Scene | 0.103 | 0.192 | 0.152 | 0.140 |
| Expression | 0.110 | 0.150 | 0.162 | 0.061 |
| Action | 0.132 | 0.057 | 0.156 | 0.158 |

Figure 3: Features extracted from movie clips in the test set by the modality-specific models. T-SNE is adopted for dimensionality reduction. The red points represent the features with label greater than 0 and the blue points represent the features with label less than 0.

valence and arousal tasks while expressions are only used in the valence task and actions are only used in the arousal task. The training process is described in section 3.3.1.

To compare the different modal combinations, we carry out several experiments with different setups, i.e. audio + scene, audio + action, audio + scene + human expression and audio + action + scene using the same training strategy. We also conduct two baseline experiments to investigate performance of different the modal fusion strategies: feature-level fusion with traditional training method, i.e. training the entire model at the same time, which is referred as Baseline 1, and decision-level fusion, i.e. learning modality-specific models independently and averaging the results, which is referred as Baseline 2. For the valence task, both baseline models use audio, scenes, and expressions. For the arousal task, we use audio, scenes, and actions. The results of various methods are listed in Table 2.

As shown in Table 2, our proposed modal selection and feature-level fusion with our residual-based progressive training algorithm get better performance compared with traditional training method and also outperforms the decision-level fusion.

### 4.2.3 PARAMETER SELECTION FOR LONG TEMPORAL FUSION

The methods to utilize the long temporal context are described in section 3.2.2 and 3.3.2, in which, the number of clips for the valence task and the decay weights $\beta$ for the arousal task are the parameters that should be determined beforehand. Here we numerically evaluate the performance under different parameter values.

Table 2: Performance of different modal combinations & feature fusions.

| | Valence | | Arousal | |
|---|---|---|---|---|
| | MSE | PCC | MSE | PCC |
| Audio+Scene | 0.091 | 0.348 | - | - |
| Audio+Action | - | - | 0.140 | 0.293 |
| Audio+Scene+Expression | **0.089** | **0.358** | - | - |
| Audio+Action+Scene | - | - | **0.138** | **0.314** |
| Baseline 1 | 0.104 | 0.284 | 0.148 | 0.257 |
| Baseline 2 | 0.090 | 0.300 | 0.142 | 0.278 |

Table 3: Performance with long temporal context. The left part shows performance for the valence task with different number of clips; the right part shows performance for the arousal task with different decay weights.

| number of clips | Valence | | decay weights $\beta$ | Arousal | |
|---|---|---|---|---|---|
| | MSE | PCC | | MSE | PCC |
| 3 | 0.073 | 0.406 | 0.96 | 0.136 | 0.400 |
| 4 | **0.071** | **0.444** | 0.97 | 0.136 | 0.409 |
| 5 | 0.072 | 0.419 | 0.98 | **0.137** | **0.419** |
| 6 | 0.077 | 0.380 | 0.99 | 0.140 | 0.427 |

For the valence task, using 4 continuous movie clips gets the best result, which is shown in the left part of Table 3. Using fewer clips can't get enough long time information while using more clips weakens the flow of information and may introduce noises by obvious emotional change between clips. For the arousal task, the best choice of the decay weight is 0.98 as shown in the right part of Table 3. The larger decay weight will pull the arousal value of every second towards the mean value resulting the higher MSE.

Table 4: Performance compared with the state-of-the-art.

| | Valence | | Arousal | |
|---|---|---|---|---|
| | MSE | PCC | MSE | PCC |
| CERTH-ITI (Batziou et al., 2018) | 0.117 | 0.098 | 0.138 | 0.054 |
| THUHCSI (Ma et al., 2018) | 0.092 | 0.305 | 0.140 | 0.087 |
| Quan et al. (2018) | 0.115 | 0.146 | 0.171 | 0.091 |
| Yi et al. (2018) | 0.090 | 0.301 | 0.136 | 0.175 |
| GLA (Sun et al., 2018) | 0.084 | 0.278 | **0.133** | 0.351 |
| Ko et al. (2018) | 0.102 | 0.114 | 0.149 | 0.083 |
| Ours | **0.071** | **0.444** | 0.137 | **0.419** |

### 4.2.4 COMPARISON WITH THE STATE-OF-THE-ART

We present our whole emotion prediction result in Table 4. It shows that our method gets significantly better performance compared with the other works. To further prove the effectiveness of our model structure and residual-based training strategy, we also carry out experiments on classification task of the LIRIS-ACCEDE dataset which is used in MediaEval 2015 (Sjöberg et al., 2015). We use the same backbone network and the same training strategy with only the last layer changed to the softmax layer for classification. The performance compared with the other works are shown in Table 5. It can be seen that our backbone network and training strategy can still outperform the state-of-art in this task, which further demonstrates the effectiveness of our extracted features and residual-based progressive training algorithm.

Table 5: Performance compared with MediaEval 2015 related works.

|  | Valence (acc) | Arousal (acc) |
|---|---|---|
| MIC-TJU (Yi et al., 2015) | 0.420 | 0.559 |
| NII-UIT (Vu Lam & Le) | 0.430 | 0.559 |
| ICL-TUM-PASSAU (Trigeorgis et al., 2015) | 0.415 | 0.557 |
| Fudan-Huawei (Dai et al., 2015) | 0.418 | 0.488 |
| TCS-ILAB (Chakraborty et al., 2015) | 0.357 | 0.490 |
| UMons (Seddati et al., 2015) | 0.373 | 0.524 |
| RFA (Mironica et al., 2015) | 0.330 | 0.450 |
| KIT (Marin Vlastelica et al., 2015) | 0.385 | 0.519 |
| Ours | **0.459** | **0.575** |

## 5 CONCLUSION AND FUTURE WORK

In this work, we propose a comprehensive video emotion impact prediction framework. For valence and arousal tasks, modalities are carefully selected through evaluating the importance of every modality to reduce noise introduced by less task-related modalities, then the pre-trained models are used to extract semantically meaningful features. Two-time-scale structures for for valence and arousal tasks are adopted to capture the shot-long temporal context. An effective and efficient residual-based progressive training algorithm is proposed. The experimental results on LIRIS-ACCEDE dataset with several comparative studies demonstrate the effectiveness of our methods. Future work might involve how to incorporate more sophisticated time dependency models of human emotion states and explore more discriminative features such as linguistic content, visual relation, etc.

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

## A  CASES ANALYSIS

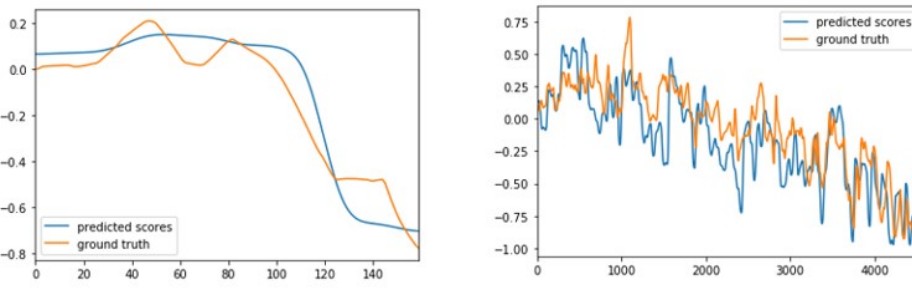

Figure 4: The predictions of movies in test set for the valence task. The left one is for MEDIAEVAL18_54; the right one is for MEDIAEVAL18_62.

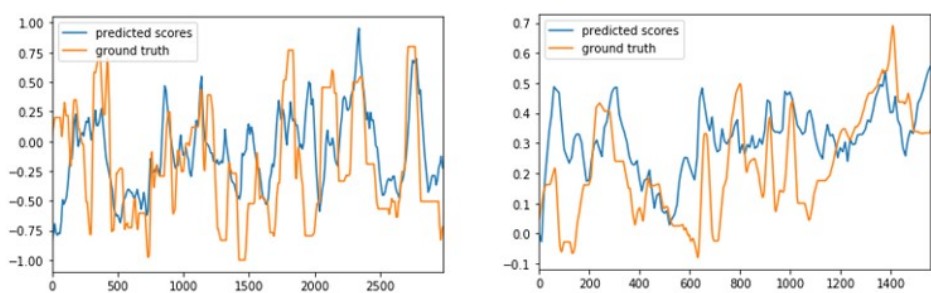

Figure 5: The predictions of movies in the test set for the arousal task. The left one is for MEDIAEVAL18_60; The right one is for MEDIAEVAL18_63.

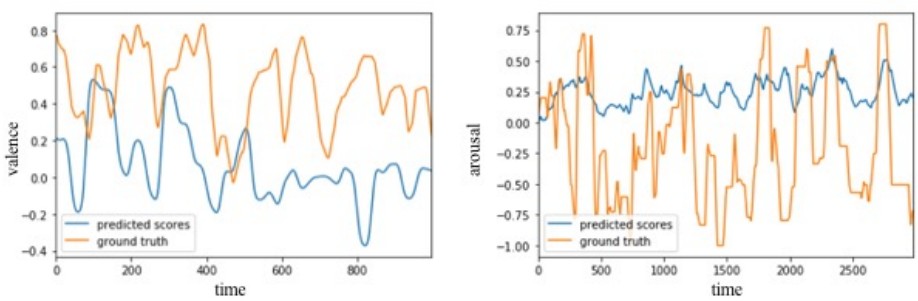

Figure 6: Bad cases in the test set. The left one is the valence predictions for MEDIAEVAL18_65 from 3000s to 4000s; The right one is the arousal predictions for MEDIAEVAL18_60.

In this part, we show both good and bad cases of the result for LIRIS-ACCEDE dataset in detail. Fig 4 and Fig 5 show the good cases, where our model can precisely predict the trend of movies' impact on audiences. Fig 6 shows some bad cases. In the left chart of Fig 6, the ground truth for most of the time are positive while the corresponding predictions are negative. This movie is a comedy, where the fighting and quarrels have positive(hilarious) emotional impact but they are difficult to identify just by the modals of our model. In the right chart, the prediction fluctuates much more than the ground truth. This is due to there are fighting and shouting in that movie clips, but the low arousal labels result from human understanding of the movie story. Those bad cases indicate the difficulty of this task since too many factors might be missing by the computer models and there is still a large gap to use machine to understand human emotions.

