# OpenReview forum: "VIDEO AFFECTIVE IMPACT PREDICTION WITH MULTIMODAL FUSION AND LONG-SHORT TEMPORAL CONTEXT"
_ICLR.cc/2020/Conference — Reject_

### Official Review · AnonReviewer3 · 2019-10-22
**Official Blind Review #3**

**Rating:** 1

**Review:**

This work design a framework to predict valence and arousal values in the emotional impact. Although its performance is much better than previous works, I have some questions about the current submission:
-- The writing is not clear enough. I have to guess the technical details based on the context. For example, in Equation 1, how to estimate y_{i}? And why put 1/m behind \sum? Is there any particular reason for this? It is a classification problem here, why use MSE loss rather than cross-entropy loss?

-- "If modality i plays an important role, it would push the mapping to complete the f_{i-1} towards f_i to get better performance." Are you suggesting that sometimes H() will be about zero?

-- Figure 2 is not clear enough for me. I still can not fully understand what ``progressive" means here. I will appreciate it if the authors can make this more clear.

-- What concerns me the most is the novelty of this work. It seems solid and effective. However, based on the current content, I can not fully appreciate its novelty. Using deep learning models to extract features from different modalities has been used before. Using deep learning to consider inter-clip and intra-clip relations also has been conducted. Using LSTM for fusion is not new, either. The authors claimed they propose a progressive training strategy for effectively training and information fusion. However, given the current version, I can not fully appreciate it.

I can change my point if the authors would provide more details and explanations later, which can help me to understand the novelty of this work fully. Thanks.

**Experience Assessment:**

I have read many papers in this area.

**Review Assessment: Checking Correctness Of Derivations And Theory:**

I assessed the sensibility of the derivations and theory.

**Review Assessment: Checking Correctness Of Experiments:**

I assessed the sensibility of the experiments.

**Review Assessment: Thoroughness In Paper Reading:**

I read the paper at least twice and used my best judgement in assessing the paper.

---

### Official Review · AnonReviewer2 · 2019-10-23
**Official Blind Review #2**

**Rating:** 1

**Review:**

This paper tackles the problem of affective impact prediction from multimodal sequences. The authors achieve state-of-the-art performance by using (i) a two-time-scale temporal feature extractor, (ii) progressive training strategy for multi-modal feature fusion, and (iii) pretraining. They divide a long video sequence evenly into several clips. The idea of applying one LSTM for intra-clips (short-time) temporal feature extraction and another LSTM for inter-clips (long-time) temporal feature extraction, resulting in two-time-scale, looks reasonable choice but not novel. The proposed progressive training strategy is mainly used for the modality-specific LSTMs training, which is used during the intra-clip short-time modeling phrase. It seems working well in practice. However, from the explanation, it's not clear why this strategy is good for the *complementary* fusion. Each modality LSTM is trained sequentially with features extracted from other fixed modality LSTMs. Also, the authors seem not explaining why they set the training order for LSTMs at this stage to be the descending order of their performance in the previous stage. In the experiments section, two weak baseline models and several previous state-of-the-art models are compared. However, enough ablation studies on the two-time-scale structure and proposed training strategy are not provided to demonstrate that the proposed method does provide complementary multi-modal feature fusion, which is claimed as a contribution.

Overall, although it achieves state-of-the-art performance on the task, none of the claimed contributions is novel or significant enough. It is a combination of existing ideas (but giving good performance). The proposed training procedure is also weak to be considered as a scientific contribution.


**Experience Assessment:**

I have read many papers in this area.

**Review Assessment: Checking Correctness Of Derivations And Theory:**

N/A

**Review Assessment: Checking Correctness Of Experiments:**

I carefully checked the experiments.

**Review Assessment: Thoroughness In Paper Reading:**

N/A

---

### Official Review · AnonReviewer1 · 2019-10-29
**Official Blind Review #1**

**Rating:** 3

**Review:**

This paper proposes a framework to predict valence and arousal tasks in videos. The framework mainly employs LSTM in a two-time-scale-structure to take multimodal inputs. In general, the proposed framework groups well-studied techniques to solve a well-known task of multimodal learning.

DNN based Multimodal learning has been heavily investigated for a long time. Numerous frameworks have been proposed with various success. Although the proposed framework is technically sound, the proposed "residual-based training strategy" and "long temporal fusion" are kind of trivial or lackluster. I can hardly identify any significant contributions that support a publication in top machine learning conferences such as ICLR.

**Experience Assessment:**

I have published one or two papers in this area.

**Review Assessment: Checking Correctness Of Derivations And Theory:**

I carefully checked the derivations and theory.

**Review Assessment: Checking Correctness Of Experiments:**

I assessed the sensibility of the experiments.

**Review Assessment: Thoroughness In Paper Reading:**

I read the paper thoroughly.

---

### Decision · Program_Chairs · 2019-12-19

**Decision:**

Reject

**Comment:**

There is no author response for this paper. The paper addresses the affective analysis of video sequences in terms of continual emotions of valence and arousal. The authors propose a multi-modal approach (combining modalities such as audio, pose estimation, basic emotions and scene analysis) and a multi-scale temporal feature extractor (to capture short and long temporal context via LSTMs) to tackle the problem. All the reviewers and AC agreed that the paper lacks (1) novelty, as the proposed approach is a combination of the existing well-studied techniques without explanations why and when this could be advantageous beyond the considered task, (2) clarity and motivation -- see R2’s and R3’s concerns and suggestions on how to improve. We hope the reviews are useful for improving the paper.